# The Impact of Environmental Information Disclosure on the Firm Value of Listed Manufacturing Firms: Evidence from China

**DOI:** 10.3390/ijerph17030916

**Published:** 2020-02-02

**Authors:** Yongliang Yang, Jin Wen, Yi Li

**Affiliations:** 1School of Economics and Management, Zhejiang Sci-Tech University, Hangzhou 310018, China; yangyliang228@163.com (Y.Y.); elainewen1031@163.com (J.W.); 2Faculty of Tourism and Culture, International United Faculty between Ningbo University and University of Angers, Ningbo University, Ningbo 315201, China; 3East China Sea Institute, Ningbo University, Ningbo 315211, Zhejiang, China; 4Center for Ecological Civilization of Yangtze River Delta, Ningbo University, Ningbo 315211, China

**Keywords:** environmental information disclosure, firm value, PSM–DID, policy implementation

## Abstract

In the last decade, the public concern over environmental problems has led to the emergence of environmental regulations in firms’ information disclosure on environmental practice, especially in some developing countries such as China. Based on a panel dataset composed of the listed manufacturing firms in China during 2006–2016, this paper uses the difference-in-differences (DID) model and the propensity score matching (PSM) method to investigate whether the Environmental Information Disclosure Measure (for Trial Implementation; EIDMT) affects the firm value. The results show that EIDMT exerts a significant impact on the listed manufacturing firms’ value. In consideration of the firm’s ownership, EIDMT plays a more important role in the firm value of non-state-owned firms than state-owned firms. Furthermore, using a PSM–DID model for eastern, central, and western China, we find that EIDMT significantly affects the firm value in eastern and western China but has little impact on central China.

## 1. Introduction

As the world’s most populous country and the fourth largest in area, China’s economy is growing at a faster rate than any major nation. However, its environmental problems are getting worse [1,2]. China’s severe environmental deterioration is not only caused by economic growth but also to an extent of imperfect environmental policies. Under this circumstance, the Chinese government makes strategic adjustments to alleviate conflicts of interest between environmental protection and economic development [3,4].

The firm contributes to the development of economy. However, it also leads to severe environmental problems. Accordingly, firms are increasingly required to be responsible for the impact of their activities on the environment [5]. A variety of environmental disclosure policies have introduced to regulate firms’ decisions and activities on the environment by affecting their corporate financial performance (CFP). According to prior studies, the impact of environmental information disclosure (EID) on CFP is still controversial. Some scholars provide evidence that there is a positive link between EID and CFP. According to Porter’s hypothesis, properly designed environmental regulation induces firms to break inefficient operational inertia and innovate, which lead to better productivity and profitability of the firms [6,7,8]. As a proxy for information transparency, EID avoids information asymmetry between managers and stakeholders. This confers competitive advantages to a firm [9,10,11,12,13]. While some studies document that there is a conflict between EID and CFP, because the pollution control expenditures can reduce firms’ margin profits [14]. In sum, many studies have explored the relation between EID and CFP, however the results are still equivocal.

We wonder whether or not the EID policy impacts firm value. Therefore, we explore the impact of EID on the firm value based on EID Measure (for Trial Implementation; EIDMT), which is the most vital EID policy in China. In this paper, we use the quasi-natural experiment to explore the net impact of EIDMT on firm value of state-owned and non-state-owned firms. Based on the difference-in-differences (DID) model and propensity score matching (PSM) model, we also discuss whether there is different results due to different regions. Moreover, we alter the dependent variable and EIDMT samples respectively to exam the robustness of the benchmark results.

In this paper, we mainly have two contributions. First, EDIMT is not only one of the most important and meaningful EID policies, but also the first normative and comprehensive departmental regulation on EID in China. This is the first paper that explores the impact of EID on the firm value, based on EIDMT. Besides, we discuss the net impact of EIDMT on the firm value with a quasi-experimental framework by applying the PSM–DID method. It avoids the endogeneity of the EID policy and diminishes selection bias of the sample firms. With various robust checks, the results are quite convincing.

The remainder of this paper is organized as follows: Section 2 presents a background, Section 3 provides a literature review, Section 4 describes the data and variable definitions as well as methods for empirical analysis, Section 5 presents empirical results and discussions, Section 6 considers the robustness check, and Section 7 draws a conclusion.

## 2. Background

In terms of environmental regulation policies, many countries have gone through the first two stages—adopting legal control alone and introducing market tools (such as tradable emission permits and sewage charge), and entered the third stage, that is, adopting environmental, society, and governance (ESG) policies to control environmental pollution. In 1989, Norsk Hydro released the world’s first environmental development report in Norway. Then, many countries and international organizations started to pay attention to ESG. A growing number of firms disclosed their environmental information, which awakened public awareness of ESG issues. Thus, how to effectively control the environmental pollution became one of the most important issues. As a vital way for the public and investors to understand the environmental protection status quo of firms, EID aroused social attention and become a great hit in theoretical researches. However, the laws and regulations related to EID were not that adequate in China. Therefore, the market effect of rewarding and punishing information disclosure cannot be well generated.

To establish and perfect the pollution source monitoring and information disclosure system, China started to adopt EID policies to regulate the pollution from firms since the 1990s. In 2003, Chinese government made regulations on the EID of the administrative departments for the first time. However, it was principled but lacking in binding and practicality. Securities Law of the People’s Republic of China (2005 Amendment) announced that if environmental information has had a significant impact on the stock’s trading price and the investor has not been informed, an interim report should be submitted. In 2008, the China Securities Regulatory Commission issued a regulation on the initial public offering (IPO) of firms in environmentally sensitive industries, stipulating that the IPO application documents of firms in environmentally sensitive industries must provide the investigation opinions of the national environmental protection agency. The China Securities Regulatory Commission will not approve an IPO application without investigation. In the same year, the ministry of environmental protection issued the EIDMT, which is a milestone in China’s legal science in EID. The EIDMT not only stipulated the definition and principle of EID, but also clarified the contents and procedural requirements of EID for firms. It is the first normative policy to regulate Chinese firms on EID, and it is also the first comprehensive departmental regulation on EID in China.

The EIDMT, by disclosing the environmental information of enterprises, gives extra “punishment” to enterprises violating environmental laws in addition to legal punishment. Firms with poor environmental performance will suffer from public boycotts, government fines and other penalties, and their development prospects are bleak. If the firm is forced to close because of environmental accidents, investors may be in a disastrous situation. Firms with strong environmental management abilities and good environmental performance have a comparative competitive advantage. From the EID by the firm, external capital providers can understand the environmental behavior and performance of the firm and make effective investment decisions. However, in existing literature, the scholars may pay more attention to the relation between EID and corporate financial performance, based on the regression method [10,15,16]. There is still no clear answer to whether and how EID affects the firm value. Considering the listed manufacturing firms disclosing environmental information as the treatment group, and the other listed manufacturing firms as the control group, we use the PSM–DID method to identify the causality between the EID and the firm value and examine the net impact of the environmental policy on the firm value.

## 3. Literature Review

### 3.1. Firm Value and Book-to-Market Ratio

The book-to-market ratio is the ratio of a stock’s book value to its market value. It is believed that stocks with high book-to-market ratios earn higher average returns than stocks with low book-to-market ratios. According to previous literature, the book-to-market ratio is always used to forecast the firm’s value by predicting stock returns. Accounting-based valuation theory asserts that the firm’s value is a combination of the book value and market expectations of future earnings. In addition, many scholars use the book-to-market ratio to explain the cross-sectional variation in stock returns [17,18,19]. Kothari and Shanken (1997) evaluate the ability of the book-to-market ratio to track time-series variation in expected market returns, and they find considerable evidence that there is a link between stock returns and the book-to-market ratio [19]. Pontiff and Schall (1998) examine whether the aggregate book-to-market ratio could forecast market returns, and they assert that the book-to-market ratio of the Dow Jones Industrial Index predicts market returns during the period of 1926–1994 [18]. Therefore, it is appropriate to use the book-to-market ratio to evaluate the firm’s value in this paper.

### 3.2. The Impact of EID on Corporate Financial Performance (CFP)

Generally, EID can positively affect CFP by inducing firms to innovate and diminishing the information asymmetry in the firms. According to the Porter hypothesis, properly designed EID policies induce firms to break inefficient operational inertia and innovate, which allows firms to improve their productivity and profitability [6,7,8]. The firms emphasize environmental disclosure can attract the investors and to fulfill the stakeholders’ demands. Therefore, more extensive environmental disclosure can confer competitive advantages to a firm. Belkaoui et al. (1976) undertake one of the first studies that documented the relations between environmental voluntary disclosure and the market response. They find that the common stock prices of disclosing corporations would be favorably affected by the disclosure of pollution control expenditures [9]. As a proxy for information transparency, EID has a significant positive impact on the firm value, such as the Tobin’s Q value, financial leverage, and return of assets (ROA) [10]. It will have a greater influence on the firm value when a firm faces high financial risks, because creditors will demand more information be disclosed to keep themselves informed as to the latest corporate developments and to make their economic decisions [11].

However, many empirical results also show a negative or no significant relation between EID and CFP. Palmer et al. (1995) assert that market competition forces firms to improve operational efficiency and innovate even without environmental regulation. In addition, the exogenous regulation affects firm profit maximization, which decreases the firm profits [14]. Hou (2018) uses corporate social responsibility (CSR) awards as an indicator of social responsibility. The study finds that socially responsible firms in Taiwan can achieve financial results superior to those of firms that do not pursue CSR initiatives [20]. Iwata (2014) analyzes the comparison between mandatory and voluntary EID and asserts that firms are likely to use more pollution abatement under mandatory EID, which has a negative effect on the firms’ profits [21]. In general, prior studies on the relation between environmental disclosure and corporate financial performance have obtained mixed results.

### 3.3. Information Disclosure’s Role in the Firm Value

Information disclosure plays a monitoring role in avoiding information asymmetry between managers and shareholders. Listed firms often use timely disclosure and annual reports as proxies for information transparency. Prior research on the information disclosure documents that the firms’ information disclosure strategies are mainly driven by three types of concerns: product market competitors [22,23,24], media [25,26], and financial analysts [12].

Product market competition has often been demonstrated as an alternative source of discipline. Sheikh (2018) uses Tobin’s Q to measure firm value, and he reports that the CEO power that is driven by product market competition has a positive effect on the firm value [27]. Many studies have documented that information disclosure has a different impact on the firm value in firms that operate in a different product competition environment. Most of scholars have reached a consistency that when product market competition is low, information disclosure has a significant positive effect on firm value [28]. Sheikh (2018) uses instrumental variable estimation regressions to examine how product market competition affects the relation between CSR and the firm value. The result shows that CSR has no impact on the firm value in firms that operate in low product competition environments or face low product fluidity, and the influence of CSR depends on CSR strength, instead of CSR concerns [29]. However, Li et al. (2019) assert that carbon information disclosure, nonfinancial carbon information disclosure, and financial carbon information disclosure are negatively correlated with the cost of equity financing [30].

The media plays a vital role in the financial market by disseminating information about firms, and firms are sensitive to the media reports by influencing the reputation capital of firms [31]. Wang et al. (2019) obtain 308 media releases related to positive environmental activities between January 2015 and December 2014 and classify the media-released information into four parts: descriptions of past actions, announcements of planned future activity, recognition for environmental stewardship, and annual reports [32]. They assess that there is a strong link between the market and announcements of planned future activity. Baloria and Heese (2017) use a quasi-natural experiment to explore the relation between media slant and firm behavior, and they document that the media can impose reputational costs on firms [33].

Firms also provide information disclosure through financial reports, which include financial statements, notes to the financial statements, and management discussion and analysis. The firm’s financial accounting quality can weaken its information risk and hence lower the firm’s cost of equity capital [34]. Zimmerman (2013) argues that external financial reporting quality has, at best, a 2nd-order effect on firm value of U.S. publicly traded firms [35].

### 3.4. Mechanism of EID’s Influencing Firm Value

Early studies examining the relation between EID and firm value often focus on the impact of specific environmental issues, such as Union Carbide’s chemical leak in Bhopal, India in 1984 [36] and Superfund Amendments and Reauthorization Act (SARA) [37]. They both find that firms with more extensive EID had a weaker negative impact on firm value. For one reason, EID can enhance firm value through improved transparency and accountability, and enhanced stakeholder trust [27]. For another reason, extensive and objective environmental disclosures can enhance a firm’s share price, as they help create a positive and strong firm reputation as well as other competitive advantages [24]. Clarkson et al. (2008) examine the impact of voluntary environmental disclosure on firm value within five sensitive industries. They document that there is a positive relation between EID and firm value. To verify whether this positive relation is due to a cash flow component, they regress future realized profitability on their disclosure measures [38].

In contrast, some scholars believe that more improved environmental disclosure leads to more precise earnings forecasts by analysts, which reduces expected future cash flow and cost of equity capital (COEC) through a reduction in information risk [12,16]. Thus, better environmental disclosure is associated with a lower firm value. For example, Badrinath and Bolster (1996) examined stock market reactions to EPA judicial actions on a sample of publicly traded firms and found that the companies suffered an average loss of 0.43% of their firm value after the environmental penalty information was disclosed [39]. There is no clear answer to whether and how EID affects the firm value. Thus, the study’s first hypothesis was that:

**Hypothesis** **1:**
*There is a statistically significant positive association between EID and the firm value.*


Many studies have documented that corporate governance mechanisms can affect the firm’s information disclosure through ownership, board mechanisms, audit committee quality, governor independence, and so on [40,41,42,43]. Besides, guanxi plays a moderating role on the relation between firm value and EID, because guanxi includes relationships with government agencies and bureaucrats [44,45]. State-owned firms are bound to take more responsibility for environmental protection because of their ownership. Therefore, they would be willing to convey their positive attitudes towards EID. Relatively speaking, non-state-owned firms are in a weak position, so they will put political strategy in a more important position. Thus, the study’s second hypothesis was that:

**Hypothesis** **2:**
*EIDMT plays a more important role in the firm value of non-state-owned firms than state-owned firms.*


## 4. Methodology and Data

### 4.1. Methodology

It is widely accepted that the DID is the best method for studying quasi-natural experiments [46]. Bertrand (2004) pointed out that one of the preconditions for the validity of the DID model is that the treatment group and the control group must meet the parallel trend test before treatment [47]. As shown in Figure 1, we could find that the curves of the treatment group and the control group of the full sample were almost parallel before 2008. However, after 2008, the firm value of the treatment group was higher than that of control group, and the gap between the two trends increased noticeably. This preliminarily proves that the DID model is a proper method to test the impact of environmental information disclosure policy on the firm value of listed companies. Therefore, the environmental information disclosure method (trial) can be regarded as a quasi-natural experiment, and the influence of policies on the performance of listed manufacturing companies can be evaluated by using the DID method. According to the principle of the DID method, we define the listed EID firms as the treatment group and the others as the control group. We constructed the following model:(1)$$y_{i,t}=\beta_{0}{+\beta }_{1}{\mathrm{treat}}_{\mathrm{it}}{+\beta }_{2}T_{t}{+\beta }_{3}{\mathrm{treat}}_{\mathrm{it}}{*T}_{t}+\sum \tau_{j}{*X}_{\mathrm{jit}}{+\varepsilon }_{i}+\delta_{\mathrm{it}}$$
where subscript i represents different listed companies; t represents time variables; y represents the firm value; treat is a dummy variable reflecting the listed EID firms, and treat=1 refers to EID firms, otherwise, treat=0 [48]; $T_{t}$ is a year dummy variable; X is the matrix vector of the control variables; δ is the coefficient matrix of control variables; and ε is the stochastic disturbance term. The estimated coefficients for four cases are shown in Table 1.

Different listed companies have heterogeneity. Therefore, when using the DID method, the listed companies, which have similar characteristics to the treatment group but do not have environmental information disclosure, should be selected as the control group. The DID method can solve the endogeneity, but it is difficult to solve the sample deviation problem, which can result in a nonrandom policy’s implementation. While the PSM method can solve the sample selection deviation. The EIDMT is an EID policy with a quasi-experimental framework. To measure the net impact of policy operation, we used the PSM method to statistically compare the sample with a diminished systematic difference based on the propensity scores [3,49,50,51].

The PSM provides the propensity score, which implies the probability of the implementation of EIDMT, which can be calculated based on the logit regression as Equation (2):(2)$$p\left(x_{b}^{l}\right)=\mathrm{pr}\left(D_{l}{=1|x}_{b}^{l}\right)=\frac{\exp\left(\lambda_{b}x_{b}^{l}\right)}{1+\exp\left(\lambda_{b}x_{b}^{l}\right)}$$
where $x_{b}^{l}$ represents the observable characteristics of the Chinese listed manufacturing firm l, which influences its EID. The propensity score $p\left(x_{b}^{l}\right)$ reflects the likelihood of firm l. $D_{l}$ is a dummy variable, and when firm l discloses its environmental information, $D_{l}=1$; otherwise, $D_{l}=0$. $\lambda_{b}$ is the corresponding coefficient. Select non-EID firms are matched to the EID firms based on their propensity scores. In addition, kernel matching uses a weighted average of non-EID firms to construct the counterfactual match for each firm, as shown in Equation (3):(3)$$\omega \left(m,n\right)=\frac{K(\frac{p_{m-}p_{n}}{R})}{\sum K(\frac{p_{m-}p_{n}}{R})}$$
where $p_{m}$ and $p_{n}$ denote the propensity scores of EID firm m and non-EID firm n. $\omega \left(m,n\right)$ is the weight used in kernel matching, and R is a bandwidth parameter.

### 4.2. Variables and Data

To comprehensively measure the impact of environmental information disclosure on the firm value, we needed to exclude the impact of other factors on the firm value. Therefore, we combined the PSM method with the DID method to solve the endogenous problem, to estimate the effects of external shocks on the impact of EIDMT. Table 2 defines the control variables we used in this paper.

Data relating to these firms were collected from the CSMAR financial and trading database, and the initial observation samples were screened according to the following four principles: (1) the manufacturing firms listed in the Shanghai stock exchange and the Shenzhen stock exchange from 2005 to 2011 were selected as the research samples; (2) because the regulatory system and statement structure of financial listed companies were quite different from those of other industries, we excluded the samples from the financial industry based on the general treatment methods of previous studies; (3) Special Treatment (ST) samples were excluded; and (4) the continuous variables were winsorized at the level of 1%.

Finally, 6066 “firm-panel” observations were grouped as unbalanced panel data, among which 5266 were labeled as the control group and 800 as the treatment group. The descriptive statistics of variables are shown in Table 3. The results of correlation coefficient and multicollinearity test are shown in Table A1 and Table A2, respectively.

## 5. Results and Discussion

### 5.1. Benchmark Regression

Chinese firms usually count on the support from local governments [53]. Therefore, we categorized listed manufacturing firms into state-owned firms and non-state-owned firms. Based on Equation (1), benchmark regression results are shown in Table 4. Columns (1), (3), and (5) were the results of regression without the control variables. The three regression results show that the coefficients of the interaction term (T×Treat) were all significantly positive at the 1% level, indicating that the firm value of the manufacturing firms publicizing the environmental information is larger than that in other manufacturing firms. The coefficient values show that the firm values of entire manufacturing firms, state-owned manufacturing firms, and non-state-owned firms in the treatment group were 0.46%, 0.31%, and 0.37% higher than those in the control group, respectively. After introducing the control variables in columns (2), (4), and (6), the coefficients of the interaction term (T × Treat) were still significantly positive, though there were some changes in the values of the coefficients. In addition, we also examined the result based on the sample from 2002 to 2016 in Table A3, the result was robust.

To overcome systematic differences in the listed manufacturing firms we selected and to reduce the bias of the DID estimation, we used the PSM method to match the EID firms with non-EID firms. First, we used both logit and probit regression to estimate the propensity scores of the listed manufacturing firms in China. Then, we selected the non-EID firms whose individual characteristics were similar to the EID firms, according to kernel matching. To avoid bias, we used the Gaussian, Biweight, Uniform, and Tricube algorithms to examine the results and the results are provided in Table A11. Finally, we evaluated the net effect of EIDMT on firm value with the PSM-DID method, the results are reported in Table 5. We found that the coefficients of the interaction term (T × Treat) were still significantly positive, which was consistent with the benchmark regression. The coefficients of the interaction terms of the entire firms and non-state-owned firms were both statistically significant at the 1% level, and the regression coefficient of state-owned firms was also significantly positive at the 5% level from logit regression, which implies that the firm values of listed manufacturing firms in China were significantly associated with the implementation of EIDMT. The result of the sample from 2002 to 2016 applying the PSM–DID method was still robust, which is presented in Table A4.

Based on Equation (3), we obtained the results of the differences of the characteristic variables before and after using the kernel matching in Table 6. Before kernel matching, there were significant differences between EID firms and non-EID firms. For example, the mean difference of cash flow between EID and non-EID firms before employing the PSM method was significant at the 1% level before the implementation of EIDMT. Namely, there was a sample selection bias. However, after the matching, the *p*-value of the cash flow after applying the PSM method was more than 10%, which indicates that there was no significant difference between the EID firms and non-EID firms, consisting with the result of the sample selected from 2002 to 2016 in Table A5.

The results of sample comparison before and after using the PSM method are reported in Table 7. According to the *p*-value, the standardized difference of the sample after matching was smaller than that before matching, which indicates that the systematic difference between the EID firms and matched non-EID firms was effectively avoided. The result based on the sample between 2002 and 2016 in Table A6 was consistent with it. Thus, the sample was supposed to be appropriate, the implementation of EIDMT could be regarded as a random experiment. The results based on the PSM–DID method were more convincing than the sole use of the DID method.

### 5.2. Region-Based Discussion

Table 8 shows the regression results for eastern, central, and western China. In terms of the coefficients of the interaction term T × Treat, EIDMT significantly affected state-owned firms in western China but showed no statistically significant influence on firm value in eastern and central China. This result indicates that, in eastern China, non-state-owned firms pay more attention to the publicity of environmental information. For state-owned firms, the estimated coefficient in western China was statistically significant at the 1% level. However, for the entire sample, the interaction terms in eastern and western China were both significant at 1% level. This result was robust, even when we used the data from 2002 to 2016 in Table A7.

### 5.3. Placebo Test

Placebo test is a kind of counterfactual test that makes contrary assumptions about the impact of a policy or event. In this paper, we conducted it to examine the robustness of the results above. We set the date of EID in advance, by multiplying the year dummy variables and Treat to establish two new interaction terms. Table 9 shows the results of regressions of the entire, state-owned, and non-state-owned firms. The coefficients of interaction terms in 2007 (yr07 × Treat) were not statistically significant, whose *p*-values were greater than the 10% level, indicating that in 2007, EIDMT did not affect the firms’ value. The result was consistent with the result based on the sample from 2002 to 2016 in Table A8.

## 6. Robustness Check

### 6.1. Alternative Dependent Variables

Although we measured the firm value by the book-to-market ratio, there are also other means to measure the firm value. Therefore, we adopted share price returns as the dependent variable, using a bootstrap self-sampling 500 times to estimate the effect of EIDMT. Since the PSM–DID method can be more convincing that it diminishes the sample selection bias, we applied it instead of the sore DID method. The results are provided in Table 10. We found that the estimated coefficients of the entire sample were still statistically significant, which indicates that there was a positive link between the environmental disclosure and the market value of the firm. This result is consistent with Blacconiere and Patten (1994). After changing the sample from 2005 to 2011 into the one from 2002 to 2016 in Table A9, the result was still robust.

### 6.2. Alternative EIDMT Samples

As discussed, our study focused on the impact of EIDMT on the firm value of listed manufacturing firms from 2006 to 2016. To provide further assurance of the results, we investigated the robustness of our results with the firms in heavy polluting industries. The results based on the PSM–DID method are shown in Table 11. The coefficients of the interaction term T × Treat were significantly positive at the 1% level in the entire sample and the non-state-owned sample, which were consistent with the aforementioned results in the manufacturing industry in Table 5. After applying the sample between 2002 and 2016 in Table A10, the result is still robust.

## 7. Conclusions

Through one of the most important EID policies in China, this study investigated the impact of EIDMT on the firm value by applying the PSM–DID method. The key conclusions were drawn as follows: (1) EIDMT exerted a significant impact on the listed manufacturing firms’ value in China. After we altered the EIDMT samples, the results show that EIDMT also had a positive effect on the firms in heavy polluting industries. A possible reason is that EID plays a monitoring role in diminishing information asymmetry so they can attract more investors. Besides, EIDMT gives extra punishment to firms with poor EID by public boycotts and government fines so firms with good environmental disclosure have a comparative competitive advantage. (2) EIDMT played a more important role in the firm value of non-state-owned firms, rather than state-owned firms. State-owned firms are bound to take more responsibility for environmental protection because of their ownership. (3) The findings’ region-based regressions proved that EIDMT significantly affected the firm value in eastern and western China but had little impact on central China. Since most of the non-state-owned listed manufacturing firms were in eastern China; the number of state-owned firms was much higher than the non-state-owned firms in western China.

The contributions of this paper to the previous literature were as follows. First, the existing researches on the relation between environmental policies and the firm value mainly concentrate on corporate social responsibility (CSR) and environmental, social, and governance (EGS) disclosure, but lacks discussions of EID. This paper is the first study focusing on the net impact of EIDMT on the firm value. Second, this paper used the PSM–DID method to provide rigorous empirical results on the net effect of EID, which avoids exogenous shocks automatically.

From the perspective of the government, we provided a rigorous reference for relevant policymaking to maximize the impact of environmental policies by considering the current situation of listed companies in China. This is also of certain guiding significance to the development of environmental protection in China. From the perspective of the firms, our contribution is to provide an empirical reference for the listed firms. We found that firms with EID had a better firm value, which gave firms an extra chance to promote their value and corporate financial performance.

There were some potential limitations in this study. First, this paper investigated whether EIDMT affects the firm value and what the impact is, but did not well discuss how EIDMT affects the firm value (Table A12). Future researches are supposed to seek the mechanism of it. Second, the data of this paper only focused on the Chinese manufacturing and heavy polluting industry. Hence, for further research, it is suggested to conduct a study among more industries, such as the chemistry, textile, and mining industry.

## Figures and Tables

**Figure 1 ijerph-17-00916-f001:**
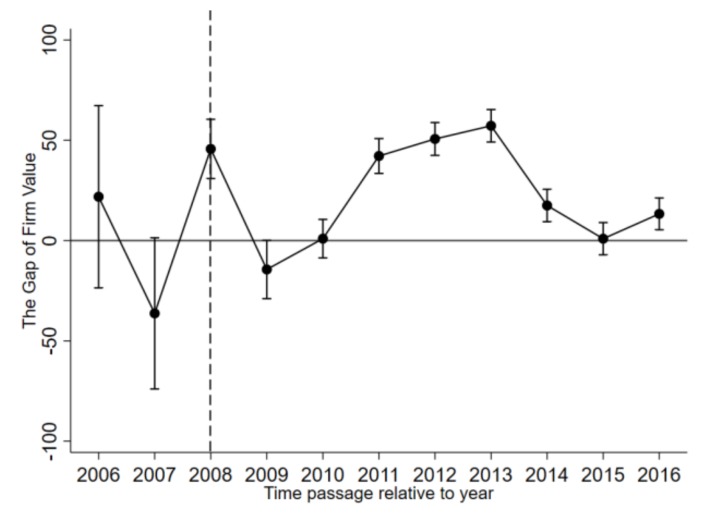
The dynamic impact of environmental information disclosure (EID) on the firm value

**Table 1 ijerph-17-00916-t001:** Meaning of each parameter in the difference-in-differences (DID) model.

Status	Before 2008 (Pt = 0)	After 2008 (Pt = 1)	Difference
Treatment group, treat = 1	$\beta_{0}{+\beta }_{1}$	$\beta_{0}{+\beta }_{1}{+\beta }_{2}{+\beta }_{3}$	$△y_{t}=\beta_{2}{+\beta }_{3}$
Control group, treat = 0	$\beta_{0}$	$\beta_{0}$+$\beta_{2}$	$△y_{0}=\beta_{2}$
Difference in difference (DID)	$\beta_{1}$	$\beta_{1}{+\beta }_{3}$	$△△{y=\beta }_{3}$

**Table 2 ijerph-17-00916-t002:** Definitions and operations of variables.

Variable Name	Variable Symbol	Definitions of Variables
Book-to-market ratio	BMR	Total assets/Market value
supervisor	supervisor	The number of supervisors
Leadership structure	LS	The dummy variable, i.e., 2 for firms whose CEO is not the COB, 1 otherwise
Percentage of top ten shareholders	PTSH	The sum of the percentages of the top ten shareholders (%)
Separation	separation	Separation of powers, difference between ownership and control
Share percentage	mshare	The sum of supervisors’, managers’, and executives’ shareholding ratios
Asset turnover ratio	turnover	Total Revenue/Total asset
Investment level	il	(Fixed assets + Construction work in process + Project material)/Last fixed asset [52]
Leverage	lev	Total assets/ Total liabilities
Sales growth rate	growth	Average real sales growth percent per year [52]
Cash ratio	cash	Cash and cash equivalents/Current liabilities
Composite tax rate	taxrate	(Business tariff and annex + Income tax expense)/Gross revenues
Year	T	The year dummy variable. 1 for years after 2008, 0 for years before 2008
Treat	Treat	The dummy variable. treat=1 refers to EID firms, 0 otherwise

Notes: BMR, LS, PTSH, and T are the abbreviations of Book-to-market ratio, Leadership structure, Percentage of top ten shareholders, and Year, respectively.

**Table 3 ijerph-17-00916-t003:** Descriptive statistics of variables.

Variable	N	Mean	sd	Min	Max
BMR	15363	0.840	0.780	0.0100	12.10
supervisor	15894	3.730	1.130	3	7
cash	15896	0.190	0.140	0.0100	0.680
taxrate	15837	0.0300	0.0300	−0.0100	0.160
PTSH	15324	58.99	15.24	22.41	92.95
turnover	14170	0.720	0.440	0.120	2.570
LS	14247	1.750	0.430	1	2
il	13402	70.16	250.8	1.570	1961
lev	15896	0.420	0.200	0.0500	0.880
growth	14169	0.180	0.350	−0.490	2.060
separation	14599	5.620	8.060	0	29.58
mshare	15324	0.0900	0.170	0	0.670

**Table 4 ijerph-17-00916-t004:** Benchmark regression results for the entire sample.

	Entire		State-Owned Firms		Non-State-Owned Firms	
	(1)	(2)	(3)	(4)	(5)	(6)
T	−0.346 ***	−0.101 ***	−0.225 ***	−0.146 ***	−0.266 ***	−0.0700 **
	(−19.37)	(−4.50)	(−5.93)	(−3.87)	(−11.54)	(−2.51)
Treat	−0.0943 *	−0.0256	0.0130	0.0184	−0.184 ***	−0.120 **
	(−1.86)	(−0.46)	(0.15)	(0.22)	(−3.38)	(−2.36)
T × Treat	0.462 ***	0.247 ***	0.310 ***	0.303 ***	0.365 ***	0.183 ***
	(6.75)	(3.52)	(2.78)	(2.89)	(5.25)	(2.79)
X	NO	YES	NO	YES	NO	YES
*N*	6066	4708	3179	2640	2554	1820

Notes: To report as complete information as possible, this paper only reports the regression results of the dummy variables omitting the other control variables and the constant term; the *t*-value is reported in the parentheses below; *, **, and *** represent the 10%, 5%, and 1% significance levels, respectively.

**Table 5 ijerph-17-00916-t005:** Regression results based on the propensity score matching (PSM)–DID method.

	Logit			Probit		
	(1)	(2)	(3)	(4)	(5)	(6)
	Entire	State-Owned	Non-State-Owned	Entire	State-Owned	Non-State-Owned
T	−0.226 ***	−0.166 ***	−0.208 ***	−0.227 ***	−0.178 ***	−0.199 ***
	(−8.33)	(−4.01)	(−6.12)	(−8.66)	(−4.40)	(−5.87)
Treat	−0.0530	0.0536	−0.190 ***	−0.0635	0.0545	−0.175 ***
	(−0.91)	(0.65)	(−3.34)	(−1.11)	(0.62)	(−3.05)
T × Treat	0.382 ***	0.262 **	0.389 ***	0.387 ***	0.264 **	0.371 ***
	(4.75)	(2.31)	(4.63)	(4.69)	(2.30)	(4.17)
X	YES	YES	YES	YES	YES	YES
*N*	4729	2813	1588	4690	2810	1615

Notes: This paper uses the logit regression to calculate the propensity score as benchmark results, and probit regression for the robust check; the *t*-value is reported in the parentheses below; *, **, and *** represent the 10%, 5%, and 1% significance levels, respectively; all control variables and the constant term are included but not reported; parameters are matched by the Epanechnikov kernel function. We also try other matching methods, and the result is robust in Appendix C.

**Table 6 ijerph-17-00916-t006:** Comparison between samples before and after applying the kernel matching algorithm.

Variable	Unmatched	Mean			*t*-Test	
	Matched	Treated	Control	% Bias	*t*	*p* > |t|
supervisor	U	3.953	3.689	22.6	11.04	0.000
	M	4.097	3.928	14.5	2.30	0.022
cash	U	0.182	0.197	−11.2	−5.08	0.000
	M	0.165	0.165	−0.3	−0.06	0.954
taxrate	U	0.027	0.026	4.6	2.24	0.025
	M	0.025	0.024	3.7	0.57	0.567
PTSH	U	58.310	59.127	−5.3	−2.51	0.012
	M	55.616	54.955	4.3	0.74	0.460
turnover	U	0.793	0.709	18.5	8.72	0.000
	M	0.829	0.832	−0.6	−0.10	0.920
LS	U	1.807	1.736	17.1	7.66	0.000
	M	1.854	1.827	6.4	1.27	0.205
il	U	54.194	73.855	−8.3	−3.55	0.000
	M	55.185	78.842	−10.0	−1.75	0.080
lev	U	0.464	0.413	25.4	11.92	0.000
	M	0.497	0.482	7.6	1.36	0.175
growth	U	0.147	0.183	−10.5	−4.64	0.000
	M	0.139	0.172	−9.6	−1.66	0.096
separation	U	6.162	5.501	8.1	3.75	0.000
	M	6.708	6.704	0.1	0.01	0.993
mshare1	U	0.056	0.092	−23.0	−10.00	0.000
	M	0.021	0.030	−6.1	−2.08	0.037

Notes: In the first column, U refers to the sample before employing the PSM, while M denotes the matched sample after applying the PSM.

**Table 7 ijerph-17-00916-t007:** Comparison between samples before and after applying the PSM method.

Sample	Pseudo-R	LR chi ^2^	*p* > chi ^2^	Mean Bias	B (%)	R	%Var
Unmatched	0.032	377.96	0.000	14.1	45.1 *	1.06	82
Matched	0.007	12.68	0.315	5.7	11.2	0.89	45

Notes: Pseudo-R implies the goodness-of-fit of logit regression; the Likelihood Ratio chi^2^ (LR chi^2^) denotes the adequacy of the logit regression; the *p*-value accesses the significance probability value; B (%) provides the standardized bias difference between the unmatched and matched samples.

**Table 8 ijerph-17-00916-t008:** Regression results for eastern, central, and western China.

Explanatory Variable	Entire			State-Owned Firms			Non-State-Owned Firms		
	(1)	(2)	(3)	(4)	(5)	(6)	(7)	(8)	(9)
	Eastern	Central	Western	Eastern	Central	Western	Eastern	Central	Western
T	−0.209 ***	−0.235 ***	−0.282 ***	−0.119 *	−0.207 ***	−0.183 **	−0.177 ***	−0.220 **	−0.381 ***
	(−6.12)	(−4.43)	(−4.57)	(−1.94)	(−2.91)	(−2.22)	(−4.66)	(−2.19)	(−4.91)
Treat	−0.0565	−0.0585	−0.131	0.0925	0.230	−0.206	−0.117*	−0.366***	0.0750
	(−0.84)	(−0.36)	(−1.11)	(0.78)	(1.04)	(−1.56)	(−1.75)	(−3.31)	(0.27)
T × Treat	0.304 ***	0.497 **	0.599 ***	0.0593	0.338	0.587 ***	0.355 ***	0.420 ***	0.319
	(3.55)	(2.20)	(3.23)	(0.41)	(1.13)	(2.82)	(3.63)	(2.59)	(0.91)
X	YES	YES	YES	YES	YES	YES	YES	YES	YES
*N*	2712	1076	789	1383	750	562	1093	233	123

Notes: the *t*-value is reported in the parentheses below; *, **, and *** represent the 10%, 5%, and 1% significance levels, respectively; all control variables and the constant term are included but reported; all parameters are estimated based on the logit regression; parameters are matched by the Epanechnikov kernel function.

**Table 9 ijerph-17-00916-t009:** Placebo test results.

	(1)	(2)	(3)	(1)	(2)	(3)
	Entire	State-Owned	Non-State-Owned	Entire	State-Owned	Non-State-Owned
yr06 × Treat	−0.332	−0.306	−0.680 ***			
	(−1.54)	(−1.24)	(−11.17)			
yr07 × Treat				−0.0951	0.0241	–0.0410
				(−1.34)	(0.24)	(−0.55)
X	YES	YES	YES	YES	YES	YES
*N*	5397	2941	2126	5436	2944	2148

Notes: the *t*-value is reported in the parentheses below; *, **, and *** represent the 10%, 5%, and 1% significance levels, respectively; all control variables and the constant term are included but reported; parameters are matched by the Epanechnikov kernel function.

**Table 10 ijerph-17-00916-t010:** The impact of EID Measure (for Trial Implementation; EIDMT) on share price returns.

	Logit			Probit		
	(1)	(2)	(3)	(4)	(5)	(6)
	Entire	State-Owned	Non-State-Owned	Entire	State-Owned	Non-State-Owned
T	−0.00213 ***	−0.00302 ***	−0.000911 *	−0.00228 ***	−0.00301 ***	−0.000935 *
	(−6.38)	(−6.82)	(−1.65)	(−6.44)	(−7.45)	(−1.65)
Treat	0.00104	0.000503	0.00191	0.00116	0.000745	0.00206
	(1.32)	(0.54)	(1.24)	(1.35)	(0.82)	(1.45)
T × Treat	0.00257 ***	0.00270 **	0.00203	0.00242 **	0.00244 **	0.00191
	(2.68)	(2.51)	(1.07)	(2.37)	(2.23)	(1.08)
X	YES	YES	YES	YES	YES	YES
*N*	4763	2818	1628	4735	2790	1614

Notes: This paper uses the logit regression to calculate the propensity score as benchmark results, and probit regression for the robust check; the *t*-value is reported in the parentheses below; *, **, and *** represent the 10%, 5%, and 1% significance levels, respectively; all control variables and the constant term are included but reported; parameters are matched by the Epanechnikov kernel function.

**Table 11 ijerph-17-00916-t011:** The impact of EIDMT on the heavy polluting industries’ firm value.

	Logit			Probit		
	(1)	(2)	(3)	(4)	(5)	(6)
	Entire	State-Owned	Non-State-Owned	Entire	State-Owned	Non-State-Owned
T	−0.225 ***	−0.134 **	−0.247 ***	−0.220 ***	−0.139 **	−0.230 ***
	(−5.78)	(−2.16)	(−4.97)	(−5.56)	(−2.33)	(−4.54)
Treat	0.0412	0.0593	−0.180 *	0.0356	0.0545	−0.153
	(0.50)	(0.60)	(−1.91)	(0.43)	(0.57)	(−1.47)
T × Treat	0.329 ***	0.238	0.522 ***	0.335 ***	0.218	0.480 ***
	(2.89)	(1.59)	(3.55)	(2.81)	(1.57)	(3.15)
X	YES	YES	YES	YES	YES	YES
*N*	2482	1629	708	2480	1617	700

Notes: This paper uses the logit regression to calculate the propensity score as benchmark results, and probit regression for the robust check; the *t*-value is reported in the parentheses below; *, **, and *** represent the 10%, 5%, and 1% significance levels, respectively; all control variables and the constant term are included but reported; parameters are matched by the Epanechnikov kernel function.

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
