# Peer review of "The Impact of Environmental Information Disclosure on the Firm Value of Listed Manufacturing Firms: Evidence from China"

_ijerph, 2020, doi:10.3390/ijerph17030916_

Round 1

Reviewer 1 Report

As it stands the paper will need some improvements before it is ready for publication. The introduction should offer information on “what”, “why”, “how” and “so what”. The author(s) do seem to address some of these questions, but in a manner which could be improved.

Introduction: please be specific about the contributions of the paper. In particular, draw on both the theoretical and empirical literature and the country context. Structure - the structure of your paper can be improved. Please re-structure as follows: (i) Introduction; (ii) Background; (iii) Theoretical literature review; (iv) Empirical literature review and hypotheses development; (v) Research design; (vi) Empirical results and discussion; and (vii) Summary and conclusion. Please refer to the papers below for guidance on how to better structure your paper. Background - please outline clearly background issues relating to the topic in the Chinese context. Highlight specific features in China that renders it interesting to situate your study on. Theoretical literature review - please identify and engage in explicit discussions of your theoretical framework in a separate section. You need a theoretical framework to adequately support the hypotheses. The paper needs both: a theoretical framework and a hypothesis or set of hypotheses. Empirical literature review and hypotheses development: for each hypothesis, please first outline the theoretical link between the variables. Second, outline the prior empirical findings. Third, present any contextual insights and finally, set up your hypotheses. Empirical results - please present them as follows. For each hypotheses, state what the findings are. Second, indicate whether the relevant hypothesis is supported or not. Third, compare and contrast the findings with those of prior theoretical and empirical studies. Fourth, highlight any implications of your study. Summary and Conclusion - please rewrite that by expanding discussions relating to findings, contributions, implications and limitations of your study.

Otherwise, I wish the author(s) well with this research.

Ntim, C.G., Soobaroyen, T. & Broad. M. J. 2017. Governance structures, voluntary disclosures and public accountability. The case of UK higher education institutions. Accounting, Auditing & Accountability Journal, 30(1), 65-118.

Al-Bassam, W.M., Ntim, C.G., Opong, K.K. & Downs, Y. 2018. Corporate Boards and Ownership Structure as Antecedents of Corporate Governance Disclosure in Saudi Arabian Publicly Listed Corporations. Business & Society. 57(2) 335-377.

Author Response

Response to comment: (As it stands the paper will need some improvements before it is ready for publication. The introduction should offer information on “what”, “why”, “how” and “so what”. The author(s) do seem to address some of these questions, but in a manner which could be improved. Introduction: please be specific about the contributions of the paper.

Response: We are very sorry for our negligence of the preciseness of expression. Thank you very much for your valuable advice. We have made correction according to your comments.We rewrite part of the introduction in Line40-74 based on your suggestion. In particular, we emphasize two contributions of this paper in Line64-70.

Response to comment: (Structure - the structure of your paper can be improved. Please re-structure as follows: (i) Introduction; (ii) Background; (iii) Theoretical literature review; (iv) Empirical literature review and hypotheses development; (v) Research design; (vi) Empirical results and discussion; and (vii) Summary and conclusion.)

Response: This comment was highly appreciated. We re-structure as follows: 1. Introduction; 2. Background; 3. Literature review; 4. Methodology and data; 5. Results and discussion; 6. Robustness check and 7. Conclusions

Response to comment: (Background - please outline clearly background issues relating to the topic in the Chinese context. Highlight specific features in China that renders it interesting to situate your study on. )

Response: This comment was highly appreciated and we have made corrections according to your comments. We have added the research background according to your opinion, which we think is very valuable. It starts with the rise of ESG in the world. Then it introduces the development of environmental information disclosure in China, the first is the requirements of government departments. Then there is the process and detailed rules of EIDMT.

Response to comment: (In particular, draw on both the theoretical and empirical literature and the country context. Theoretical literature review-please identify and engage in explicit discussions of your theoretical framework in a separate section. You need a theoretical framework to adequately support the hypotheses. The paper needs both: a theoretical framework and a hypothesis or set of hypotheses. Empirical literature review and hypotheses development: for each hypothesis, please first outline the theoretical link between the variables. Second, outline the prior empirical findings. Third, present any contextual insights and finally, set up your hypotheses. )

Response: Although your proposal that divide the literature review into the theoretical and empirical literature is very innovative, but we haven't finished the change yet. A very important reason is that the research of influence channel is the direction of our future research, which needs to be further deepened. This paper is a concise empirical paper, so the literature we quoted is mainly empirical paper. Our focus is on the net effect of policy, not the process of policy influence. This proposal is very good, and we will adopt this structure in further research in the future. Your point of view is very meaningful and we have added the Hypothesis 1 in L266 and H2 in L276.

Response to comment: (Empirical results - please present them as follows. For each hypotheses, state what the findings are. Second, indicate whether the relevant hypothesis is supported or not. Third, compare and contrast the findings with those of prior theoretical and empirical studies. Fourth, highlight any implications of your study.)

Response: This comment was highly appreciated and we have re-written this part according to the Reviewer’s suggestion. There is a little difference in the presentation structure of the results, because we want to show the robustness of the results of this method.

Response to comment: (Summary and Conclusion - please rewrite that by expanding discussions relating to findings, contributions, implications and limitations of your study. Otherwise, I wish the author(s) well with this research.)

Response: We reorganized the conclusion section according to your suggestion.

Response to comment: (Please refer to the papers below for guidance on how to better structure your paper. Ntim, C.G., Soobaroyen, T. & Broad. M. J. 2017. Governance structures, voluntary disclosures and public accountability. The case of UK higher education institutions. Accounting, Auditing & Accountability Journal, 30(1), 65-118. Al-Bassam, W.M., Ntim, C.G., Opong, K.K. & Downs, Y. 2018. Corporate Boards and Ownership Structure as Antecedents of Corporate Governance Disclosure in Saudi Arabian Publicly Listed Corporations. Business & Society. 57(2) 335-377.)

Response: We have read these two articles carefully, which has benefited us a lot. According to the articles, this paper makes some modification and reasonable reference. The reason why the structure of these two articles is not fully adopted is because of the differences in research problems and methods.

Once again, thank you very much for your good comments and suggestions.

Reviewer 2 Report

Dear Authors, I believe your paper is well designed and the results are clear. In terms of the methodology I think you used is sound. I have no comments on the research methodology, the results and the conclusions. However, in my humble opinion, the research does not adds much to the existing literature. I find the research question superficial. I do not learn as much as I would like given the very interesting database you have.

Particularly because the EIDMT is compulsory. I bet there is output data of emissions and the like that could be more beneficial for the research to understand whether EIDMT has actually benefited the environment over the firms impact.

Author Response

Response to comment: (Dear Authors, I believe your paper is well designed and the results are clear. In terms of the methodology I think you used is sound. I have no comments on the research methodology, the results and the conclusions. )

Response: Thank you very much for your approval and encouragement. We have further improved the method to make the result more rigorous and credible.

Response to comment: (However, in my humble opinion, the research does not adds much to the existing literature. I find the research question superficial. I do not learn as much as I would like given the very interesting database you have.)

Response: Your evaluation is pertinent and objective. In the last draft, we did not express our contribution to the existing research. In this revised version, we have made great improvement on this issue in L64-70 and L518-523. We have also made some improvements to the literature review, so that this paper can be better integrated into the current research。

Response to comment: (Particularly because the EIDMT is compulsory. I bet there is output data of emissions and the like that could be more beneficial for the research to understand whether EIDMT has actually benefited the environment over the firms impact.)

Response: This point of view is of great value. Although EIDMT is mandatory for heavy polluting enterprises, it has little impact on non-heavy polluting enterprises. It's really more meaningful to study emissions, but the data on corporate emissions are very difficult to obtain in China. Through sorting out the environmental disclosure information of enterprises, we find that most of the information disclosed is soft information, and the description of enterprise pollution emission lacks specific values. We are also making more attempts and hope to finish the research work of pollution discharge as soon as possible.

Once again, thank you very much for your good comments and suggestions.

Reviewer 3 Report

This paper have investigated whether EIDMT affects firm value by using the difference-in-differences (DID) model and the propensity score matching (PSM) method. My comments are as follows.

(1)In the table 2, authors should define all variables including treat and T.

(2) In the table 3, authors should show descriptive statistics of all variables. The data of some variables have some outliers, such as PTSH and il, which may affect the results of regressions. Authors should winsorize all continuous variables at the 1% and 99% levels. The PTSH is the sum of the percentages of the top ten shareholders, however, its max value is 101.8 more than 1,why?

(3)Authors have compared the change of firms value pre- and post-EIDMT from 2006-2016, and set T=0 before 2008(2006-2008,three years) and T=1 after 2008(2009-2016, eight years). However, we often compare the same years before and after a shock, such as 3 years data before and after a shock. So this paper may have a comparability problem. From the figure 1, it is found that firms value has decreased in 2009 and 2010 comparing that of 2008, which is not consistent to the regressions’ results.

(4) EIDMT is implemented in 2008 and all firms are affected by EIDMT, so it is not a good exogenous event because there is no controls sample which is not affected by the EIDMT after 2008. Some robust checks should be added such as 2SLS regressions.

(5) It is difficult to understand the logit and probit regression in table 10 and 11 because the dependent variables are share price returns and firm value.

(6) Authors can’t tell us how the EIDMT affects the firms value. In general, EIDMT should affect the firms value by the cash flow and expected return rate.

Author Response

Response to comment: (1)In the table 2, authors should define all variables including treat and T.

Response: Your opinion is very helpful and we are very sorry for our negligence of the preciseness of expression in the table 2. We have added information according to your suggestion.

Response to comment: (2) In the table 3, authors should show descriptive statistics of all variables. The data of some variables have some outliers, such as PTSH and il, which may affect the results of regressions. Authors should winsorize all continuous variables at the 1% and 99% levels. The PTSH is the sum of the percentages of the top ten shareholders, however, its max value is 101.8 more than 1, why?

Response: Thank you very much for your valuable advice. We have added descriptive statistics of all variables in Table 3. We winsorize all continuous variables at the 1% and 99% levels in Table 3.

We also report benchmark regression results for the sample winsorized at 1% level in Table B1. The result is robust. I'm sorry we missed the dimension, which we added % in Definitions of variables this time. The PTSH of some companies is abnormal due to cross shareholding. Because of the complex equity structure, the CSMAR database we use does not calculate the final net shares.

Response to comment: (3)Authors have compared the change of firms value pre- and post-EIDMT from 2006-2016, and set T=0 before 2008(2006-2008,three years) and T=1 after 2008(2009-2016, eight years). However, we often compare the same years before and after a shock, such as 3 years data before and after a shock. So this paper may have a comparability problem. From the figure 1, it is found that firms value has decreased in 2009 and 2010 comparing that of 2008, which is not consistent to the regressions’ results.

Response: The opinions given are of great significance and give us new ideas. With regard to figure 1, I have to apologize. In the past, we marked the y-axis as Firm Value, but this is misleading, which we change to The Gap of Firm Value. As for the problem that the value in 2008 may be larger than that in 2009 and 2010, I think we should not pay attention to the local level, because 2008 is the highest value in the previous three years, but should notice the trend. The result of DID is also tested by strict hypothesis in Table 4. Your advice is very helpful and we report benchmark regression results for the sample three-year pre- and post-EIDMT in Table B2. The results are consistent with the whole sample although there are some changes in the size and significance of the coefficients.

Response to comment: (4) EIDMT is implemented in 2008 and all firms are affected by EIDMT, so it is not a good exogenous event because there is no controls sample which is not affected by the EIDMT after 2008. Some robust checks should be added such as 2SLS regressions.

Response: The reviewer is precise and rigorous and this problem really needs us more attention. Because EIDMT requires that heavy polluting enterprises should be made public, and other industries should be encouraged to be made public, so the difference of policy strength among industries is obvious. We think that our choice of DID method is suitable for the current research problems. For 2SLS, we tried many potential variables, but did not find the ideal IV. We have also tried to use GMM, but this method consumes a lot of sample capacity. Before 2009, there were not many samples in the treatment group, and GMM results are not stable here. It's a pity that we can't do more about it.

Response to comment: (5) It is difficult to understand the logit and probit regression in table 10 and 11 because the dependent variables are share price returns and firm value.

Response: You are very friendly and remind us of our shortcomings. I'm sorry for our negligence, because we didn't give enough explanation. This paper uses the logit regression to calculate the propensity score as benchmark results, and probit regression for the robust check. Now we explained it in Notes below the table.

Response to comment: (6) Authors can’t tell us how the EIDMT affects the firms value. In general, EIDMT should affect the firms value by the cash flow and expected return rate.

Response: Some channels about the influence of EIDMT on firm value are provided in the current literature, so we added 3.4 of literature review in Line236-262. This paper focuses on estimating the net effect of policy differences. We do have some shortcomings in explaining this phenomenon, which we think is the direction of further research in the future.

Once again, thank you very much for your good comments and suggestions.

Reviewer 4 Report

The research presented in this paper is indeed very useful and will be of interest to a wide range of academics and practitioners given the growing importance of environmental information disclosure 

Some minor concerns:

1) Make clear the overall contribution of the analysis. Please, put more emphasis on the novelty of the paper and the theoretical/practical implication

2)Some literature review sections is rough and please provide some HP.

See for example Lagasio and Cucari (2019). https://doi.org/10.1002/csr.1716 

Cucari, N., Esposito De Falco, S., & Orlando, B. (2018). Diversity of board of directors and environmental social governance: Evidence from Italian listed companies. Corporate Social Responsibility and Environmental Management25(3), 250-266.

3)  Improve the discussion of the paper by providing a clear linkage between theoretical background and HP result.

Author Response

Response to comment: (The research presented in this paper is indeed very useful and will be of interest to a wide range of academics and practitioners given the growing importance of environmental information disclosure.)

Response: Thank you very much for your affirmation.

Response to comment: (Some minor concerns: 1) Make clear the overall contribution of the analysis. Please, put more emphasis on the novelty of the paper and the theoretical/practical implication)

Response: The suggestion has been taken seriously. We rewrite part of the introduction in Line40-74 based on your suggestion. In particular, we emphasize two contributions of this paper in Line64-70. We have added the research background in Line75-130 according to your opinion, which we think is very valuable. It starts with the rise of ESG in the world. Then it introduces the development of environmental information disclosure in China, the first is the requirements of government departments. Then there is the process and detailed rules of EIDMT. We also reorganized the conclusion in which there are theoretical/practical implications.

Response to comment: (2)Some literature review sections is rough and please provide some HP. See for example Lagasio and Cucari (2019). https://doi.org/10.1002/csr.1716. Cucari, N., Esposito De Falco, S., & Orlando, B. (2018). Diversity of board of directors and environmental social governance: Evidence from Italian listed companies. Corporate Social Responsibility and Environmental Management, 25(3), 250-266. )

Response: This comment was highly appreciated. We have made some corrections according to your comments. We have read these two articles carefully, which has benefited us a lot. According to the articles, this paper makes some modification and reasonable reference. The reason why the structure of these two articles is not fully adopted is because of the differences in research problems and methods. Some channels about the influence of EIDMT on firm value are provided in the current literature, so we added 3.4 of literature review in Line236-262. Your point of view is very meaningful and we have added the Hypothesis 1 in Line266 and H2 in Line276. We reorganized the expression of literature review in Line156-159, Line 167-172, Line208-213, Line 231-234.

Response to comment: (3)  Improve the discussion of the paper by providing a clear linkage between theoretical background and HP result.)

Response: We reorganized the expression of in Line490-517 and more discussion is added in Line518-535.

Once again, thank you very much for your good comments and suggestions.

Round 2

Reviewer 3 Report

Some questions are not resolved perfectly.

(1) Authors should winsorize all continuous variables at the 1% and 99% levels and all regressions should use the data winsorized.

(2) I insist that authors should use the same years pre- and post- EIDMT in paper, instead in Appendix.

(3)It is confused that in table B2, both T and Treat have the negative relationship with firm value, but T×Treat has the positive effect on firm value. This results may have some abnormal. From Table 1. Correlation coefficients, I can’t see the relationship of T and BHR, and I guess it is negative. This is not consistent to this paper’s conclusions.

(4)Authors should consider the exogenous problems.
